# Complexation Nanoarchitectonics of Carbon Dots with Doxorubicin toward Photodynamic Anti-Cancer Therapy

**DOI:** 10.3390/jfb13040219

**Published:** 2022-11-05

**Authors:** Thu Thi Anh Do, Kukuh Wicaksono, Andree Soendoro, Toyoko Imae, María José Garcia-Celma, Santiago Grijalvo

**Affiliations:** 1Graduate Institute of Applied Science and Technology, National Taiwan University of Science and Technology, 43 Section 4, Keelung Road, Taipei 10607, Taiwan; 2Department of Chemical Engineering, National Taiwan University of Science and Technology, 43 Section 4, Keelung Road, Taipei 10607, Taiwan; 3Department of Pharmacy, Pharmaceutical Technology, and Physical-Chemistry, IN2UB, R+D Associated Unit to CSIC, Pharmaceutical Nanotechnology, University of Barcelona, Joan XXIII 27-31, 08028 Barcelona, Spain; 4Networking Research Center on Bioengineering, Biomaterials and Nanomedicine (CIBER-BBN), Jordi Girona 18-26, 08034 Barcelona, Spain

**Keywords:** nitrogen-doping carbon dot, oxygen photosensitization, singlet-oxygen generation, drug delivery, photodynamic therapy

## Abstract

Carbon dots (Cdots) are known as photosensitizers in which the nitrogen doping is able to improve the oxygen-photosensitization performance and singlet-oxygen generation. Herein, the characteristics of nanoconjugates of nitrogen-doped Cdots and doxorubicin were compared with the property of nitrogen-doped Cdots alone. The investigation was performed for the evaluation of pH-dependent zeta potential, quantum yield, photosensitization efficiency and singlet-oxygen generation, besides spectroscopy (UV-visible absorption and fluorescence spectra) and cytotoxicity on cancer model (HeLa cells). Encapsulation efficiency, drug loading, and drug release without and with light irradiation were also carried out. These investigations were always pursued under the comparison among different nitrogen amounts (ethylenediamine/citric acid = 1–5) in Cdots, and some characteristics strongly depended on nitrogen amounts in Cdots. For instance, surface charge, UV-visible absorbance, emission intensity, quantum yield, photosensitization efficiency and singlet-oxygen generation were most effective at ethylenediamine/citric acid = 4. Moreover, strong conjugation of DOX to Cdots via π-π stacking and electrostatic interactions resulted in a high carrier efficiency and an effective drug loading and release. The results suggested that nitrogen-doped Cdots can be considered promising candidates to be used in a combination therapy involving photodynamic and anticancer strategies under the mutual effect with DOX.

## 1. Introduction

Cancer is the second leading cause of death in all countries worldwide [1]. The disease diagnosis and management have gained many improvements; however, the traditional cancer therapies, especially, surgery, chemotherapy or radiotherapy retain some limitations, e.g., long-term morbidity, drug resistance, and side effects [2]. Photodynamic therapy (PDT) was recently developed for various types of cancer and macular degeneration. PDT is a non-invasive treatment that involves the use of a photosensitizer that generates reactive oxygen species (ROS) under the light of a specific wavelength, causing cell damage, cell apoptosis, or necrosis [3]. A photosensitizer should possess several features to be a potential candidate for PDT, such as (i) easiness to synthesize; (ii) non-toxicity in the absence of a photostimulus; (iii) having energy of an excited state higher than the energy of singlet oxygen (0.97 eV); (iv) rapid elimination from the body; and (v) showing minimal aggregation and high photostability. Several photosensitizers for PDT are organic dyes, such as porphyrins and their derivatives, but these materials have some problems for clinical application, such as low solubility and aggregation in aqueous media, which reduce the quantum yield and make intravenous injection difficult. Conventional photosensitizers also have low selectivity and hence may affect healthy cells [4]. Recently, the combination of chemotherapy and PDT has attracted attention because it can enhance the efficiency of killing cancer cells by a synergistic effect of both treatments [5]. For example, Wang et al. [6] developed a ROS-sensitive thioketal linkage as a linker between upconversion nanoparticles and doxorubicin (DOX) for PDT + chemotherapy combination cancer treatment. Their results showed that the produced singlet oxygen can cleave the thioketal linkage, inducing DOX release and improving chemotherapy. Compared with monotherapy, the combination of two therapies enhances drug loading and effectiveness.

Carbon dots (Cdots) are promising fluorescent nanomaterials with nanometer size, high optical absorptivity, chemical stability, water dispersibility, and biocompatibility [7]. Due to their characteristic properties, Cdots can be used for a variety of applications in biosensing, optoelectronics, photocatalysis, bioimaging (in vitro and in vivo), and drug-delivery systems [8,9]. There is a wide variety of top-down and bottom-up methods for producing Cdots. For the top-down routes, chemical oxidation, discharge, electrochemical oxidation, and ultrasonic wave are mainly used to disintegrate larger structures, but they require expensive materials, harsh reaction conditions, and time-consuming techniques [10]. The bottom-up routes mostly use hydrothermal or solvothermal methods [11] besides microwave-assisted-carbonization [12] and thermal decomposition [13]. On bioimaging applications of Cdots, using their fluorescence property, Cdots have been used as green fluorescent probes for imaging HeLa cells [11]. Dually emissive P,N-co-doped Cdots with both green (430 nm) and red (500 nm) fluorescence were also used as contrast agents for photoacoustic and fluorescent imaging of mice tumors [14]. Cdots can also act as vehicles for drug- or gene-delivery systems [9]. Since small-size Cdots (i.e., below 9 nm) can not only cross the blood–brain barrier, but also diffuse freely into the cell nucleus, they can be applied either as drug-delivery vehicles or as in vivo molecular biomarkers [15]. While having tunable optical properties and fluorescence emission, Cdots also show much lower toxicity in the human body in contrast with heavy metal-semiconductor quantum dots [16]. Cdots prepared by a hydrothermal method using citric acid (CA) and ethylenediamine (EDA) and conjugated to DOX have shown higher cellular uptake and better anti-tumor efficacy on MCF-7 cells, compared to free DOX [7]. Additionally, for Cdots synthesized by a hydrothermal method from *o*-phenylenediamine and CA, DOX was efficiently loaded through electrostatic interactions, released from these complexes after 6 h incubation, and entered the cell nuclei of L929 and HeLa cells after 8 h [17]. Moreover, it has been reported that Cdots can be used as a promising candidate for not only PDT, but also photothermal therapy (PTT) due to the ability to transform absorbed light into heat [18,19].

The present study describes the synthesis of Cdots with different levels of nitrogen-doping by the hydrothermal method using CA and EDA for combination therapy. The prepared Cdots of a small size (below 5 nm) have a graphitic structure and vast numbers of functional groups, such as -NH_2_, -OH, or -COOH, which enable the conjugation of different therapeutic agents through π-π stacking, electrostatic interaction or covalent bonding [20]. The anthracene ring and amino groups of DOX can bind onto Cdots to form Cdots/DOX complexes by π-π stacking and electrostatic interactions, respectively. The high amine doping of Cdots imparts positive charges to Cdots under physiological conditions which may enhance the permeability of the cell membrane [21]. The oxygen photosensitization performance and the singlet-oxygen generation of Cdots were examined by 3,3′,5,5′-tetramethylbenzidine (TMB) and anthracene tests. Furthermore, we exploited Cdots/DOX complexes as a drug-delivery system, and then the loading and release efficiency were determined. Finally, we studied the cytotoxicity in HeLa cells to evaluate the anti-tumor activity of Cdots/DOX complexes. These investigations will prove the suitability of Cdots/DOX in combination therapy and the important role of nitrogen doped in Cdots.

## 2. Experimental

### 2.1. Materials

DOX was purchased from Sigma-Aldrich, USA. Methanol (99%), anthracene (99%), CA, and EDA were purchased from Acros Organics, UK. TMB was obtained from Dojindo, Japan. Other reagents were used in commercial-grade form. Protoporphyrin IX (PPIX) was purchased from Tokyo Chemical Industrial Co., Tokyo, Japan. Deionized water was obtained from a Millipore purification system (Yamato Millipore WT100, Japan).

### 2.2. Characterization

FT-IR absorption spectra were collected in the 400 to 4000 cm^−1^ range on a Nicolet^TM^ FT-IR instrument (Thermo Scientific, Waltham, MA, USA) using KBr pellets. Zeta potentials were measured with an ELS-Z from Photal Osaka Electronics, Japan, for dispersions (1 mg/mL) in water or phosphate buffer solution (PBS). Ultraviolet (UV)-visible absorption spectroscopic measurements were carried out on a Jasco V-670 spectrometer (Tokyo, Japan) using a 1 mm quartz cell. Fluorescence was measured with a Hitachi F-7000 fluorometer at a scan rate of 2400 nm/min using a 1 cm quartz cell. High-resolution transmission electron microscope (HRTEM) images were captured with a Philips FEI Tecnai 20 G2 S-Twin microscope at 200 kV. X-ray photoelectron spectroscope (XPS, VG Scientific ESCALAB 250, London, UK) analyses were performed to determine the chemical composition of Cdots. X-ray diffraction (XRD) was obtained by a Rigaku D/Max-Ka diffractometer (Tokyo, Japan) with CuKα (λ = 1.54184 Å) at 40 kV and 30 mA. Scattered radiation was recorded in the range of 2θ = 10–80°.

### 2.3. Synthesis of Cdots

Cdots were prepared by a hydrothermal procedure as previously described [22]. CA (1 g) was dissolved in water (50 mL) under stirring, and then EDA (300 µL) was added. The mixture was heated in an autoclave at 230 °C for 5 h. Then, the mixture was cooled down to room temperature and dried to obtain a dark brown powder. Similarly, Cdots with different molar ratios of CA:EDA, from 1:1 to 1:5, were synthesized using different EDA amounts, and the resulting Cdots were labeled as Cdots1:x, where x is the EDA amount relative to CA. 

### 2.4. Loading and release of DOX on Cdots

An aqueous DOX solution (100 μg·mL^−1^, 2 mL) was added to an aqueous Cdots1:x solution (1 mg·mL^−1^, 1 mL), and then the mixed solution was stirred at 25 °C for 48 h in the dark. The obtained Cdots1:x/DOX complexes were purified by dialyzing (6000 Da dialysis membrane) the resulting solution against 20 mL water for 6 h and were stored at 4 °C for further use.

The encapsulated efficiency and loading efficiency of DOX were spectroscopically calculated [6]: optical absorption of the supernatant was measured at the fixed wavelength of 480 nm to quantitatively obtain the concentration of free DOX based on a calibration curve. Then,
(1)$$\mathrm{Encapsulated}\;\mathrm{efficiency}\;(\%)=\frac{\mathrm{Total}\;\mathrm{DOX}-\mathrm{Free}\;\mathrm{DOX}}{\mathrm{Total}\;\mathrm{DOX}}\times 100$$
(2)$$\mathrm{Loading}\;\mathrm{efficiency}\;(\%)=\frac{\mathrm{Total}\;\mathrm{DOX}-\mathrm{Free}\;\mathrm{DOX}}{\mathrm{Cdots}\;\mathrm{mass}}\times 100$$

The drug release from the Cdots1:x/DOX complexes was studied as follows: Cdots1:x/DOX (2 mL) complexes were placed in a dialysis bag (6 kDa) and immersed in a receptor solution (PBS, 20 mL) at pH 5.6 and 7.4, under constant stirring at the physiological temperature of 37 °C. The receptor medium (4 mL) was withdrawn at selected time intervals and replaced with fresh PBS (4 mL). The DOX release (%) was quantified by fluorescence intensity at the 490 nm wavelength, according to the following formula [23]:(3)$$\mathrm{DOX}\;\mathrm{release}\;\left(\%\right)=\frac{\mathrm{Drug}\;\mathrm{in}\;\mathrm{release}\;\mathrm{solution}}{\mathrm{Total}\;\mathrm{amount}\;\mathrm{of}\;\mathrm{drug}}\times 100$$

To evaluate the DOX release for PDT, Cdots1:x/DOX (2 mL) complexes in the dialysis bags (MWCO 6 kDa) were immersed in a PBS receptor solution (20 mL) at pH 5.6 under constant stirring, and then the complexes were subjected to LED light (450 nm, 0.08 Wcm^−2^, SkyFire (LumiTorch), Taiwan) for 5 min. Aliquots of PBS were collected at selected time intervals to quantify the DOX release as described above.

### 2.5. Measurements of Quantum Yield, Oxygen Photosensitization and Singlet Oxygen Generation

The quantum yield Ф_X_ of Cdots1:x solutions was measured by using anthracene in absolute ethanol as a standard (Ф_ST_ = 0.27) [24], according to the following equation:(4)$${Ф}_{X}\;=\;{Ф}_{\mathrm{ST}}\times \frac{{\mathrm{Grad}}_{X}}{{\mathrm{Grad}}_{\mathrm{ST}}}\times \frac{{ŋ}_{X}^{2}}{{ŋ}_{\mathrm{ST}}^{2}}$$
where Grad is the gradient, and ŋ is the refractive index of solvent. The subscript “ST” and “X” refer to reference fluorophore and Cdots, respectively. 

The oxygen photosensitization performances were determined with TMB as a ROS probe. When colorless TMB is oxidized by ROS generated from photosensitization, a resulting blue-colored product is evaluated by the variation of UV-visible absorbance at 655 nm [25]. A mixture of TMB (200 μg·mL^−1^) and test samples (Cdots1:x and PPIX as a control, 10 μg·mL^−1^) was diluted with 2 mL water. Then, the mixture was irradiated under LED (450 nm, 0.08 W cm^−2^) for 20 min before measuring the absorbance.

To quantify the singlet-oxygen generation, the emission spectra of the standard fluorescence probe, anthracene, which was quenched upon oxidation by singlet oxygen, were recorded at an excitation wavelength of 320 nm [26]. Anthracene in methanol (20 μg mL^−1^, 1 mL) was mixed with test samples at different concentrations (2.5–15 μg mL^−1^, 1 mL). The mixtures were irradiated with a LED lamp (450 nm, 0.08 W cm^−2^) for 30 min before the measurement of emission spectra. The fluorescence intensity was integrated from 350 to 550 nm.

The generated singlet oxygen can be calculated by the following equation:(5)$${}^{1}O_{2}=\frac{I_{0,\mathrm{probe}}-I_{t,\mathrm{probe}}}{{\mathrm{Grad}}_{\mathrm{probe}}}\times \frac{V}{{\mathrm{MW}}_{\mathrm{probe}}}$$
where I_0_ and I_t_ are the integrated intensities of the fluorescence emission of probe before irradiation and at irradiation time t, respectively, V is the volume of a solution, and Grad_probe_ and MW_probe_ (178.23 g/mol) are the emission intensity gradient, and the molecular weight of the probe, anthracene.

### 2.6. In Vitro Cell Cytotoxicity Tests 

The cytotoxicity of Cdots1:x/DOX complexes was evaluated in HeLa cells by the MTT (3-(4,5-dimethylethiazol-2-yl)-2,5-diphenyltetrazolium bromide) assay. In brief, cells were seeded in DMEM (Dulbecco’s modified Eagle’s medium) using 96-well plates at a density of 4000 cells per well and cultured at 37 °C with 5% CO_2_ for 24 h. Then, Cdots1:x/DOX complexes at varying concentrations (0.002 to 4 µg·mL^−1^) were added to the corresponding wells, and cells were further incubated at the same condition for 48 h. After that, cells were removed and replaced by DMEM, and continuously incubated for 2 h. The cells were then subjected to the MTT assay. MTT (15 µL, 0.5 mg·mL^−1^ in PBS) was added to each well and incubated for 2 h following the change from yellow MTT into dark-blue formazan crystals. Thereafter, the medium was removed, and dimethyl sulfoxide (200 µL) was added to each well to dissolve the formazan crystals. The absorbance of the resulting cell dispersions was measured at 560 nm by using a microplate reader (Labsystem, Multiskan, Ascent, Model 354, city, Finland). The percentage of cell viability was calculated and compared to the negative control in PBS. The MTT assay was also performed to study the cytotoxicity of reference materials to HeLa cells under pH 7.4. The IC_50_ values were determined by GraphPad Prism Software. 

## 3. Results and Discussion

### 3.1. Characterization of Cdots/DOX

The strategy for the preparation and the mechanism of action of Cdots/DOX complexes are shown in Figure 1A. The Cdots series were synthesized by a hydrothermal method from CA and EDA precursors, where the molar ratio of CA:EDA was from 1:1 to 1:5. DOX binds to Cdots to form Cdots/DOX complexes by π-π stacking and electrostatic interactions. Cdots/DOX complexes enter the cells by diffusion and endocytosis [27,28]. In the acidic environment of lysosomes, DOX is released and then enters into cell nuclei. Cdots can also produce ROS and/or singlet oxygen to kill the cell via PDT [29].

The FT-IR absorption spectra of the Cdots1:x series (Figure 1B-a) showed a broad band ranging from 3000 to 3700 cm^−1^ that is assigned to O-H and N-H stretching vibration modes. Bands located at 1653, 1215, and 1048 cm^−1^ are assigned to C=O and C-O-C stretching and C-O-H bending vibration modes, respectively. N-H, C=C and C-N stretching vibration modes were allocated at 1554, 1477 and 1384 cm^−1^, respectively [30,31]. These results suggest that the Cdots possess graphitic, carbonyl, nitrogen-containing, and hydroxyl groups [22,30]. With an increasing amount of EDA, the band intensity of the N-H bending vibration mode increased, indicating the increase in nitrogen doping. A FT-IR spectrum of DOX (Figure 1B-b) showed bands at 3448, 1729, 1621, 1579, 1405, 1282, 1214, and 1021 cm^−1^. Bands at 1729 and 1621 cm^−1^ were assigned to -C=O stretching vibration modes of quinone and ketone carbonyls of DOX, respectively. In comparison with DOX, the FT-IR spectrum of Cdots1:2/DOX complex showed the characteristic bands of DOX besides bands of Cdots, illustrating the successful binding of DOX onto Cdots [7]. 

As seen in Figure 1C, XRD patterns of Cdots displayed a peak at 23°. Although this peak corresponds to the (002) graphite lattice, it was broad because of the very small size of Cdots [17]. The such small particle size of Cdots1:1 and Cdots1:1.5 were characterized by HRTEM as shown in Figure 1D, which showed uniform Cdots with around 1 nm [32]. However, Cdots/DOX complexes formed aggregates with a size range from 5 to 10 nm depending on the mole ratio of CA:EDA (Figure 1E-1 and Appendix A). It is suggested that the size increase in Cdots1:x/DOX complexes might relate to the increased electrostatic binding between carboxylic acid of Cdots and amine of DOX, resulting in increasing self-assembly and aggregation [33]. Figure 1E-2 is a HRTEM of Cdots/DOX complex. Some crystal domains were observed in a particle, and their d-spacing was 0.241 ± 0.05 nm, which corresponds to a (001) spacing of Cdots [34], indicating the combination of Cdots with DOX.

XPS full survey scans of Cdots1:1 and Cdots1:1.5 powders are shown in Figure 2A. XPS fine peaks and deconvoluted peaks of each element are shown in Figure 2B,C, and Appendix A summarizes the binding energy of deconvoluted peaks and their assignments [22,31,35]. XPS full survey scans revealed that Cdots are mostly made up of C1s, N1s and O1s located at 284.6, 399.4 and 32.5 eV, respectively. The deconvolution of C1s indicates the existence of four different carbon species, namely aromatic C-C/C=C bonds located at 284.8 eV, C-OOH/C-NH_2_ bonds centered at 285.3 eV, alkyl C-C and hydroxyl C-OH bonds located at 287.5 and 288.6 eV, respectively. An XPS fine spectrum of N1s showed one peak at 399.9 eV, associated with -NH_2_ (amine) bond. The calculated atomic percentage of nitrogen in Cdots1:1 was 9.9 %, which was lower than that of Cdots1:1.5 (12.8 %). These numerical values were close to calculated values (11.1 and 14.9 atom%, respectively). The deconvolution of O1s resulted in two peaks located at 532.2 and 533.1 eV, corresponding to C=O and C-OH bonds of carboxylic acid. 

The zeta-potential of Cdots relates to the reversible protonation and deprotonation of functional groups (amine and carboxylic acid). For instance, the amine groups of Cdots were strongly protonated in acid conditions resulting in positive zeta-potential values; however, they became neutral due to their deprotonation of amine groups when pH increased [36]. Meanwhile, carboxylic acid was neutral at acid conditions but negatively charged at alkaline conditions due to their deprotonation. As a balance of inverse behaviors of amine and carboxylic acid groups, the isoelectric point was elevated, such as pH 5.51, 6.61, 6.98, 7.44, and 8.62 from Cdots1:1 to Cdots1:5, respectively, as seen in Figure 3Aa, because of the heightening amine doping. These results suggest that the pH-dependent charge reversal is strongly related to amine doping in Cdots. As seen in Figure 3Ab, after DOX-conjugation, the zeta-potential values of Cdots1:x/DOX were similar to those of DOX but higher by about three times than those of Cdots1:x at the whole pH region because of the addition of charged DOX. Although the isoelectric points of Cdots1:1/DOX-Cdots1:4/DOX (Cdots1:1/DOX: 6.86, Cdots1:2/DOX: 8.02, Cdots1:3/DOX: 7.65 and Cdots1:4/DOX: 7.48) heightened compared to relevant Cdots1:x, they were comparable to an isoelectric point (8.50, see Appendix A) of DOX. However, the isoelectric point (pH 8.70) of Cdots1:5/DOX coincided with both Cdots1:5 and DOX. These results indicate that the zeta potential of Cdots/DOX is dependent on that of DOX and the electrostatic interaction between Cdots and DOX. 

The bioimaging of Cdots and Cdots/DOX can be characterized according to their optical properties. Therefore, Cdots and Cdots/DOX complexes were investigated by UV-visible absorption and fluorescence spectroscopies. Polar groups from precursors (carboxylic and hydroxyl groups from CA, and amine groups from EDA) are exposed in Cdots, and the products become dispersible in water, as seen in Figure 3B, where the color of the dispersion lightened with an increasing amount of amine. Figure 3C shows the Cdots1:x in aqueous media under fluorescence light exposure. Cdots emitted blue light, and the emission became brighter as the EDA-doping increased. However, after DOX conjugation, the brightness of the emitting blue light was reduced, although it was still brighter than free DOX, as seen in Figure 3D. 

These fluorescence behaviors were quantitatively examined using fluorescence spectroscopy. As seen in Figure 3E, Cdots1:2 showed a UV-visible absorption band at 240 and 350 nm that can be assigned to the π-π* transition of an aromatic skeleton and C-C in sp^2^ carbon network in the graphitic core and to the *n*–π* transition of C=O, C-N, or C-OH bonds. Then the maximum excitation and emission wavelengths of Cdots1:2 were at 360 and 465 nm, respectively. Table 1 lists the UV-visible absorption bands and fluorescence emission and excitation bands of Cdots1:x based on the UV-visible absorption and fluorescence spectra in Appendix A. With an increasing amount of EDA from 1:1 to 1:2, the excitation and emission wavelengths shifted to a higher wavelength, although such a shift could not be found at the EDA increase from 1:2 to 1:5. This red shift might be related to the variation in electric states of the Cdots [37]. On the other hand, the fluorescence intensity increased with the EDA content, but maximized at CA:EDA of 1:4.

The UV-visible absorption and fluorescence spectra of free DOX in Appendix A showed the characteristic absorption bands at 240, 260, 290, 480 and 540 nm and an emission band at 430 nm for the excitation wavelength of 340 nm. DOX also showed a characteristic emission wavelength at 560 nm for excitation wavelength 490 nm. Two fluorescence bands can be assigned to the π-π* transition of an aromatic skeleton and to the *n*–π* transition of C=O or C-OH surface bonds, and these two states are not passivated [38]. However, the UV-visible absorption spectra of Cdots1:x after loading DOX were like those of Cdots at a lower wavelength, but were like those of DOX at a higher wavelength, when Appendix A were compared with Figure 3e. Then, the emission band and its intensity of Cdots1:x/DOX intermediate compared to those of Cdots1:x and DOX, as seen in Figure 3f and Appendix A. Emission bands of Cdots1:x/DOX at excitation wavelength of 490 nm are shown in Appendix A, and Table 1 lists fluorescence bands and their intensities of Cdots1:x/DOX. The bands stayed unchanged, but the main band was intensified, and the weak band was weakened with increasing the content of EDA. 

From the emission spectra of both Cdots1:x and Cdots1:x/DOX, quantum yields were calculated and plotted in Figure 3g in comparison with the variation of fluorescence intensities. When the molar ratio of CA:EDA increased from 1:1 to 1:5, the quantum yield of Cdots gradually increased, and the highest quantum yield at 0.40 was obtained as the molar ratio was 1:4. This result qualitatively agreed with a previous report in which Cdots were prepared by a hydrothermal method using different molar ratios of L-ascorbic acid and urea (from 1:1 to 1:8) [19,37]. These results indicate that the nitrogen doping in Cdots plays an important role in their optical properties. After loading DOX, the quantum yields slightly decreased due to quenching by DOX, suggesting the π-π stacking interaction of the aromatic group of DOX with the graphitic structure of Cdots. These tendencies were seen alongside the behaviors of emission intensities. 

The photostability and photobleaching resistance of Cdots were also examined. On the photostability test for 0–1 M NaCl concentration, the fluorescence emission intensities of Cdots scarcely changed, as seen in Figure 4A. The fluorescence emission intensity of aqueous Cdots solutions with pH ranging from 3 to 13 revealed significant changes as varying pH values (Figure 4B). Under acidic conditions (pH 3 or 5), the fluorescence emission intensity of Cdots reached the highest values, but it showed fluorescence quenching with increasing pH values (pH 7–13). These results suggest that the protonation of amine in Cdots may influence the fluorescence. Cdots with protonated amines are dispersed as isolated species at low pH values and might appear as the agglomeration of Cdots at higher pH values, inducing the decrease in fluorescence. Moreover, the fluorescence intensity of Cdots has no obvious changes under 1-year storage at room temperature, as seen in Figure 4c–g. For the determination of photobleaching resistance, the fluorescence emission intensity of aqueous Cdots solutions in a phosphate buffer was recorded. The normalized fluorescence emission intensity showed the slight variation of the emission intensity with 1 h irradiation, as seen in Figure 4H, suggesting that Cdots are resistant to photobleaching. These results indicate that Cdots can become a promising candidate for biological applications. 

### 3.2. Oxygen Photosensitization Property and Singlet Oxygen Generation Performance of Cdots/DOX

The oxygen photosensitization properties of the series of Cdots1:x and Cdots1:x/DOX were investigated by using a chromogenic TMB probe and a PPIX reference [25]. The generation of a characteristic absorption band at 655 nm provides evidence that TMB is readily oxidized by ROS, as formularized in Appendix A and shown in Figure 5A, where aqueous solutions of PPIX and Cdots1:x were blue, confirming their oxygen photosensitization after LED irradiation. Figure 5b indicates that the capability of Cdots1:x and Cdots1:x/DOX to generate ROS was better than that of PPIX, the capability of Cdots1:x is always higher than that of Cdots1:x/DOX, and the EDA content at 1:4 CA:EDA in Cdots was the highest ROS generation.

Singlet oxygen is considered an important active species in ROS to cause cell death in PDT and to be detected using anthracene as a fluorescent probe, as described in Appendix A, where anthracene reacts with singlet oxygen and immediately produces non-fluorescent anthraquinone [26]. Then, the singlet-oxygen generation can be obtained from the decrease in the fluorescent signal (emission) of anthracene by the aid of Appendix A, where Cdots1:x or Cdots1:x/DOX in anthracene solution were irradiated with LED for 30 min (see Figure 5c,d). When the concentration of Cdots1:x and the amount of amine in Cdots1:x increased, the singlet-oxygen generation also increased. In the present case, a CA:EDA of the 1:4 molar ratio was the most effective for singlet-oxygen generation. Cdots1:x/DOX complexes also showed similarly the singlet-oxygen-generation effect, although their effect was less than that of Cdots1:x. These results indicate that the singlet-oxygen-generation efficiency mostly displayed similar behavior to oxygen photosensitization, meaning that main active species in ROS may be a singlet oxygen. 

### 3.3. Drug Loading and Release Response

The encapsulation efficiency and the drug-loading capacity using DOX as an anti-cancer chemotherapeutic drug model were examined at different mol ratios of CA and EDA. As seen in Figure 6A, the highest values (61.0% and 35.8%, respectively) for encapsulation efficiency and drug-loading capacity of the series were obtained for Cdots1:3. In the release profile of DOX from Cdots1:x/DOX in an RBS receptor solution at pH 5.6 (the endosomal pH of cancer cells) and 7.4 (the physiological pH) (Figure 6B), DOX showed a fast release during the initial hours, and higher drug release was obtained at pH 5.6 as compared to pH 7.4, and converged after long hours for all Cdots1:x/DOX studied. The drug release percentages after 48 h were plotted as a function of pH for different CA:EDA ratios in Figure 6C. A slight difference could be observed as a function of the CA:EDA mole ratio with a minimum release at 1:2 ratio. 

The efficiency of Cdots1:x/DOX in the production of ROS and singlet oxygen was confirmed above. ROS and singlet oxygen could damage cellular constituents through reacting with biological molecules, which plays a vital role in inducing cell apoptosis in PDT. Separately, it is known the photothermal effect of Cdots under light irradiation [18,19]. Therefore, the release of DOX within 600 min incubation under light irradiation was studied at pH 5.6 and compared with the release without irradiation as shown in Figure 6D. The results indicate that the total DOX released under light irradiation was higher than the drug released in the absence of light irradiation. In Figure 6C, the drug released from Cdots1:x under light irradiation at pH 5.6 after 600 min in comparison with that of Cdots1:x without irradiation is plotted as a function of the CA:EDA ratio. The drug release at 600 min was justifiably smaller than the release at 48 h, but the dependency on the CA:EDA ratio was similar. However, the drug release under light irradiation was fairly (1.5–2.0 times) higher than that without irradiation, increasing slightly with the EDA concentration and reaching a maximum of 83 %, even after 600 min incubation in contrast to the value of 65% achieved after 50 h without light irradiation. These results suggest that Cdots generated the photothermal effect. That is, Cdots play a role in increasing the environmental temperature using photoenergy. When the generated temperature reaches cell-damage temperature (>43 °C), cancer cells are killed. In the case of DOX doping, the release of DOX under the light irradiation is greater than that in the absence of irradiation because the interaction of Cdots and DOX are broken up by a heat-induced disruption of the electrostatic and/or π-π interaction. The released DOX will attack cancer cells [39,40]. 

In association with the effect of nitrogen doped in Cdots on singlet-oxygen production and drug loading, Markovic et al. [41] have suggested that the defects and free radicals on the surface of Cdots can promote the transfer of light energy from Cdots to oxygen, resulting in the generation of reactive oxygen species, such as singlet oxygen. Wu et al. [25] suggested that the nitrogen doped in Cdots exists as three species of graphitic nitrogen, pyridinic nitrogen, and pyrrolic nitrogen. Theoretical calculation indicates that graphitic nitrogen and pyrrolic nitrogen play important roles for triplet activation and oxygen adsorption in oxygen photosensitization and singlet-oxygen generation. However, the excess existence of nitrogen species causes self-quenching and loses fluorescence and singlet-oxygen generation as a case of Cdots1:5. 

### 3.4. In Vitro Cell Cytotoxicity Tests

For in vitro tests, HeLa cells were treated with Cdots1:x/DOX and free DOX as a control, and the MTT assay was used to measure the cytotoxicity. To evaluate the effect of DOX on the in vitro cell culture, HeLa cells were treated with several DOX concentrations bounded in Cdots1:x, ranging from 0.002 to 4 µg·mL^−1^. As seen in Figure 7A, the viability of HeLa cells was over 95% at a DOX dose of 0.002 µg·mL^−1^, and it gradually decreased below 20% as the concentration of DOX increased. A similar tendency was also observed for all Cdots1:x/DOX, although the viability was lower for Cdots1:2/DOX than for Cdots1:1/DOX, but it increased with the increase from Cdots1:2/DOX to Cdots1:5/DOX. Both Cdots1:x/DOX complexes and free DOX caused dosage-dependent death of Hela cells after 48 h incubation [37].

According to the half maximal inhibitory concentration (IC_50_), the IC_50_ value for free DOX was 0.078 µg·mL^−1^ (seen in Figure 7B). Meanwhile, the IC_50_ values of Cdots1:1/DOX, Cdots1:2/DOX, Cdots1:3/DOX, Cdots1:/4/DOX, and Cdots1:5/DOX complexes were 0.765, 0.031, 0.114, 0.104, and 0.243 µg·mL^−1^, respectively, which were similar to the tendency estimated from Figure 7A and higher than free DOX, except Cdots1:2/DOX, as listed in Appendix A. The outcomes of this series might be related not only to the charge exerted on the nanoparticles caused by a nitrogen-doping increase, but also to the efficacy of the nanocarrier in releasing the therapeutic drug into cancer cells. Thus, zeta-potential values confirmed an increase in the charge of the Cdots1:x as the nitrogen-doping levels increased at acid pH, suggesting that the positively charged surface of Cdots/DOX complexes might interact with the negatively charged cell membranes through electrostatic interactions and affect cancer cells in their growth [42,43], and the positive charge of Cdots/DOX complexes depends on the x value in Cdot1:x/DOX as known from the zeta-potential values in Figure 3Ab. Some works in the literature have reported that although the case of lower IC_50_ of DOX than of Cdots/DOX occurred for MGC-803 cells and the inverse case was true for MCF-7 cells, their IC_50_ was always larger than that of the present report for HeLa cells (see Appendix A) [7,44,45]. These tendencies may also be reasoned by the strong electrostatic interaction of Cdots/DOX with HeLa cells.

On the contrary, Cdots1:1/DOX displayed the lowest zeta-potential value and consequently was the least efficient in all concentrations used. Despite the fact that the rest of the nanoconjugates displayed positive surface charge values, only Cdots1:2/DOX showed the highest toxicity profile on HeLa cells of the series, achieving better anti-cancer effects than free DOX at low concentrations (0.002 and 0.008 µg·mL^−1^). Curiously, Cdots1:2/DOX was able to release the cytotoxic drug in a more sustainable way both at the physiological and acid pH than their Cdots counterparts, as observed in Figure 6b. This behavior might be also observed in the cell culture because of the acidic extracellular environment presented in HeLa cells, which may trigger complex dissociation and promote the release of free active DOX into cancer cells [46,47].

## 4. Conclusions

Nitrogen-doped Cdots were successfully prepared by the hydrothermal method from CA and EDA. Cdots possess excellent optical properties and high quantum yields by the advantage of nitrogen doped in Cdots, which is an indication of the importance of nitrogen doping in Cdots. Chemotherapeutic drug DOX was combined with Cdots via π-π stacking and electrostatic interactions. The variation of zeta potentials, diameters, and optical properties confirmed the attachment of DOX onto the Cdots, and they were highest at Cdots1:4. The optimal drug loading and encapsulated efficiency of DOX were highest for Cdots1:3. Meanwhile, the drug releasing profile of Cdots1:x/DOX had pH-sensitive properties, due to the better solubility of DOX in acidic conditions. In addition, Cdots under light irradiation produced ROS and singlet oxygen. These results proved the action of Cdots as a PDT agent and a trigger of the complex dissociation between Cdots and DOX, resulting in the drug release. The cell viability results suggested that Cdots1:2/DOX showed the most enhanced cytotoxicity in HeLa cells compared to free DOX at the lowest DOX concentrations. Therefore, the development of carbon-based nanomaterials might open new strategies in cancer therapy and PDT. However further in vivo experiments will be designed to confirm the feasibility of Cdots/DOX as a potential photosensitizer for dual therapy in order to improve anti-tumor properties. 

## Figures and Tables

**Figure 1 jfb-13-00219-f001:**
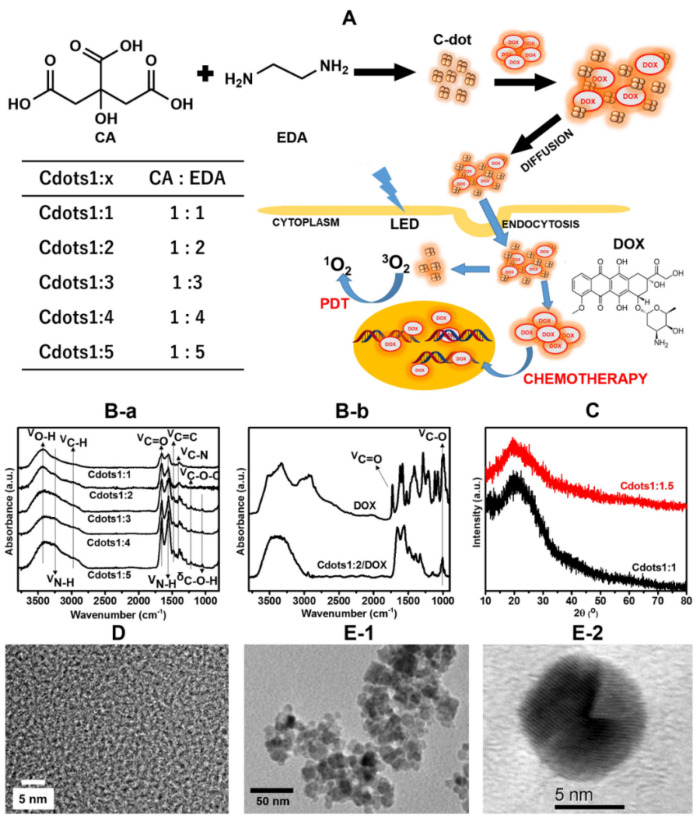
(**A**) Scheme of preparation of Cdots and Cdots/DOX complexes and combination therapy using Cdots and Cdots/DOX complexes against tumor cells. Inset: Table of the preparation of Cdots1:x series, where 1:x is the mole ratio of CA and EDA. (**B**) FT-IR spectra of (**a**) Cdots1:x and (**b**) DOX and Cdots1:2/DOX, (**C**) XRD pattern of Cdots1:x, (**D**) HRTEM of Cdots1:1.5 and (**E**) (**1**) TEM and (**2**) HRTEM of Cdots1:4/DOX.

**Figure 2 jfb-13-00219-f002:**
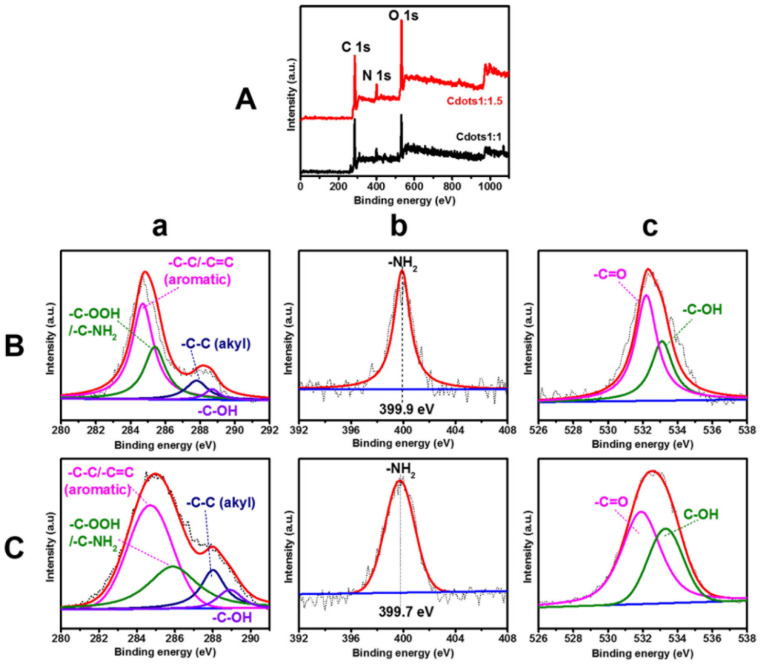
(**A**) Full-scan XPS spectra of Cdots, and fine and deconvoluted XPS spectra of (**B**) Cdots1:1, (**C**) Cdots1:1.5. (**a**) C1s, (**b**) N1s and (**c**) O1s.

**Figure 3 jfb-13-00219-f003:**
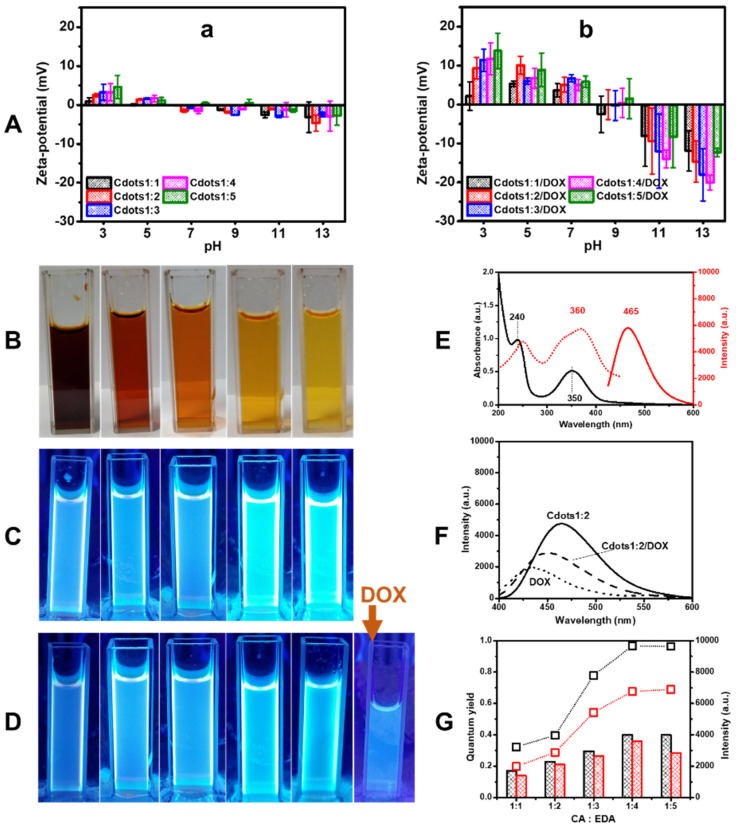
(**A**) Zeta potentials of (**a**) Cdots1:x, and (**b**) Cdots1:x/DOX. Photographs of (**B**) Cdots1:x in water (1 mg/mL, 3 mL), (**C**) Cdots1:x in water (1 mg/mL, 3 mL) under fluorescence light, (**D**) Cdots1:x/DOX in water under fluorescence light (left to right in (**C**) and (**D**): Cdots1:1 to Cdots1:5 except DOX in (**D**)), (**E**) an UV-visible absorption spectrum (black solid line) and fluorescence excitation (red dotted line) and emission (red solid line) spectra of Cdots1:2, (**F**) emission spectra of Cdots1:2 (solid line), Cdots1:2/DOX (dashed line), DOX (dotted line) at excitation wavelength 340 nm, and (**G**) quantum yield (column) and fluorescence intensity (line and symbol) of Cdots1:x (black) and Cdots1:x/DOX (red) at various CA:EDA ratios.

**Figure 4 jfb-13-00219-f004:**
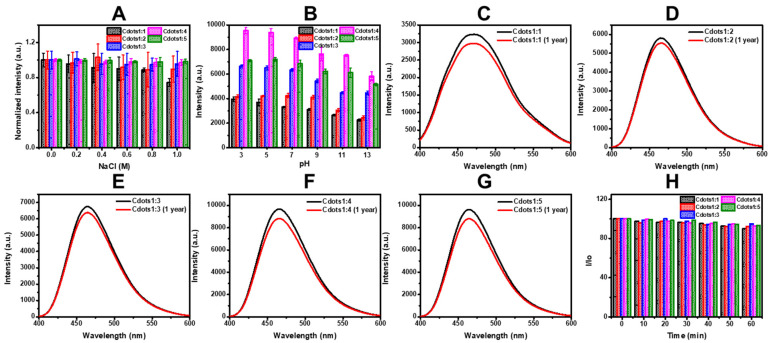
(**A**) Influence of NaCl (0.5 mL) on emission intensity of Cdots (1 mg/mL, 0.5 mL in water), (**B**) influence of pH on emission intensity of Cdots, (**C**–**G**) emission spectra of fresh Cdots and Cdots (1 mg/mL) after 1-year storage at room temperature, and (**H**) resistance to photobleaching (fluorescence emission intensity of Cdots (1 mg/mL, 1 mL in water) in phosphate buffer (50 mM, pH 7.5) for 1 h irradiation at 360 nm excitation by a 150 W quartz–xenon lamps).

**Figure 5 jfb-13-00219-f005:**
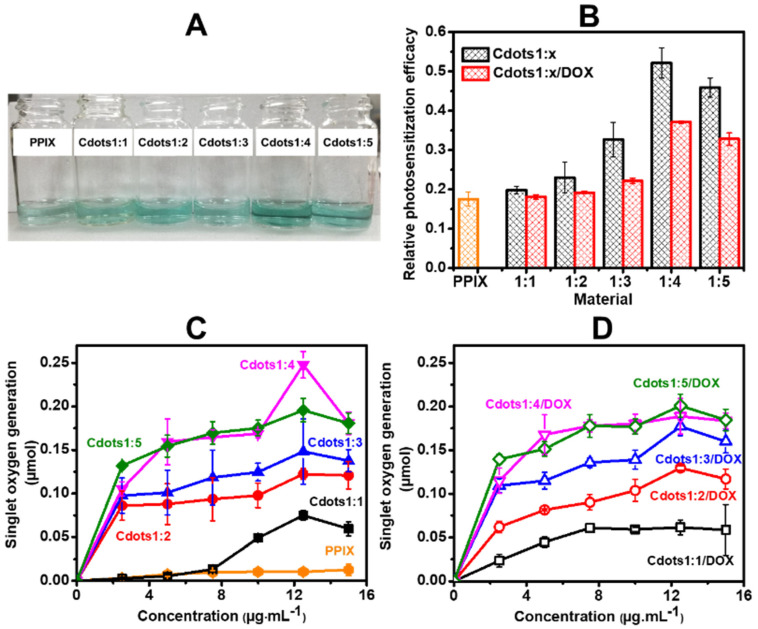
(**A**) Photographs of Cdots1:x in TMB solution after LED irradiation, (**B**) oxygen photosensitization efficiency of PPIX, Cdots1:x (black) and Cdots1:x/DOX (red) calculated from absorbance at 655 nm and singlet oxygen generation of (**C**) PPIX, Cdots1:x and (**D**) Cdots1:x/DOX. The original data of (**B**–**D**) are shown in Appendix A.

**Figure 6 jfb-13-00219-f006:**
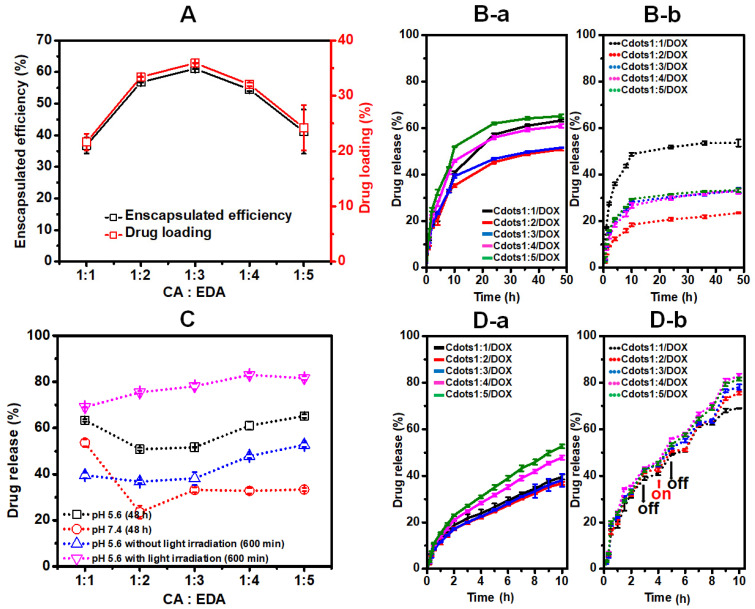
(**A**) Encapsulation efficiency and drug loading of Cdots1:x, release profiles of Cdots1:x/DOX in PBS buffer (**B**) without light irradiation at (**a**) pH 5.6 and (**b**) pH 7.4, (**C**) drug release (%) plot of Cdots1:x/DOX as a function of CA:EDA in different conditions, and (**D**) at pH 5.6 (**a**) without light irradiation and (**b**) with light irradiation.

**Figure 7 jfb-13-00219-f007:**
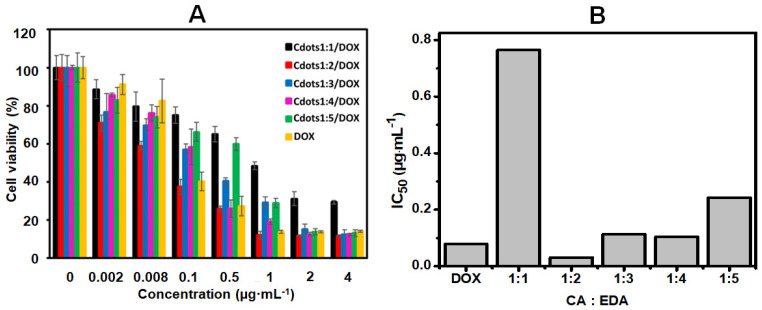
(**A**) In vitro cell cytotoxicity test and (**B**) IC_50_ plot as a function of CA:EDA.

**Table 1 jfb-13-00219-t001:** UV-visible absorption bands and fluorescence emission and excitation bands of Cdots1:x, Cdots1:x/DOX and DOX.

	UV-Visible	Fluorescence		
	Absorption Band (nm)	Excitation Band (nm)	Emission Band (nm)	Intensity (a.u.)
Cdots1:1	235, 340		471	3234
Cdots1:2	240, 350	360	465	5810
Cdots1:3	240, 350	360	465	6760
Cdots1:4	240, 350	360	466	9672
Cdots1:5	240, 350	360	464	9633
Cdots1:1/DOX	235, 340, 450, 514	340, 490	441, 542	1977, 584
Cdots1:2/DOX	240, 360, 451, 530	340, 490	452, 551	2865, 466
Cdots1:3/DOX	240, 360, 450, 530	340, 490	442, 547	5359, 396
Cdots1:4/DOX	240, 360, 490, 580	340, 490	441, 548	6703, 368
Cdots1:5/DOX	240, 360, 490, 580	340, 490	442, 549	6869, 303
DOX	230, 260, 290, 480, 540	340, 490	430, 560	1949, 803

## Data Availability

All data have been made available through the manuscript.

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
