# Peer review of "Complexation Nanoarchitectonics of Carbon Dots with Doxorubicin toward Photodynamic Anti-Cancer Therapy"

_jfb, 2022, doi:10.3390/jfb13040219_

Round 1

Reviewer 1 Report

This manuscript describes the Cdots1:X with excellent optical properties and high quantum yields by the advantage of nitrogen doped in Cdots. Chemotherapeutic drug DOX was combined with Cdots via π-π stacking and electrostatic interactions, resulted in a high carrier efficiency and an effective drug loading and release. Then, the author founded that the variation of zeta-potentials, diameters, and optical properties confirmed the attachment of DOX onto the Cdots, and they were highest at Cdots1:4. The optimal drug loading and encapsulated efficiency of DOX were highest for Cdots1:3. The cell viability results suggested that Cdots1:2/DOX showed most enhanced cytotoxicity in HeLa cells than free DOX at the lowest DOX concentrations. Overall, although the study is well designed and overall quality and presented data are solid, the data have several problems that need to be addressed:

1.     It is recommended that the authors should optimize the charts in the article, since the text of the charts in the article is not clear and the graphics are fuzzy. The quantitative analysis is missing in Fig.7A

2.     The uppercase and lowercase letters of the charts in the article are not uniform. For example, the letter labels in Fig. 4 are lowercase, while the letter labels in other pictures are uppercase. The author should unify them.

3.     How to prove the conjugation of DOX to Cdots is π-π stacking and electrostatic interactions.

4.     The authors mentioned in the text Cdots/DOX complexes enter the cells by diffusion and endocytosis, but the author did not to confirm.

5.     According to the TEM, Cdots are prone to aggregate which is not favored by biomedical applications. The PDI and DLS of Cdots and Cdots1:X/DOX should be given.

6.     As far as I know, DOX is not stable under light conditions, so will the light conditions have effect on DOX in some experiments that require lighting?

7.     The cytotoxicity of Cdots and Cdots1:X/DOX to mammalian normal cells needs to be tested.

8.     The results showed that light irradiation can facilitate the release of DOX and generate singlet oxygen. Therefore, the authors are encouraged to study the cytotoxicity of Cdots1:X/DOX upon light irradiation.

9.     To state that the combination of PDT and DOX is a promising strategy in this study, more experiments should be carried out to illustrate the synergistic effect of this system, and the synergy coefficient should be given.

Author Response

Attached file.

Reviewer 2 Report

This manuscript by Toyoko Imae and coworkers studied several nanoconjugates of nitrogen-doping carbon dot and doxorubicin. The surface charge, UV-visible absorbance, emission intensity, zeta-potentials, quantum yield, photosensitization efficiency and singlet oxygen generation were systematically studied. The experimental data support the conclusion well. The results are interesting. My comments are:

For the cytotoxicity study, I think that LED light should be shine to the cells to compare with the dark condition to prove that the PDT really happened.

Page 8 line 203, is there any publication support that DOX is released and then enters into cell nuclei? Can the authors cite the papers?

Page 5 line 118, I think there is a typo 1 mm quartz cell should be 1 cm. 

Author Response

Attached file.

Reviewer 3 Report

The optical properties study of these system presents several flow which make this paper not suitable at this time for publication.

The method to mesure singlet oxygen should be enhanced. And more experimental conditions should be given for luminescence experiments (such as optical density). Authors should explain why doxorubine features several emissions wavelengths.

Table 1, figures S3 and figure 3. The DOX is absorbing at and above 500 nm, therefore, I don’t understand how it is possible to see fluorescence at 450 nm for DOX alone in figuez 3. A pure molecule should have only one emission peak. The presence of two emission peaks suggests that two different species are present in the sample. As DOX pocesses several acidic protons, variuos emission bands means that the molecule exist in different protonation states in the sample.

Page 4. L. 115. Zeta-potentials were measured with an ELS-Z from Photal Osaka Electronics, Japan, for dispersions à  Zeta-potentials were measured on an ELS-Z from Otsuka Electronics, Japan, for dispersions.

In page 7 In the equation for singlet oxygen generation, I am not convenced by this equation because :

1)      This  equation gives 1O2 in L.g-1.mol.

2)       I don’t understand what is Grad standard. Standard is the anthracene or other photosensitizer used as a standard ?

3)      I0-It is the decrepancy of the fluorescence of anthracene during the experiment ?   

Page 9. L. 230 There is no carboxylic acids in doxuribine, only phenol and alcohols groups.

In figure S1. The zeta potential of DOX is presented. However, DOX is a molecule, therefore I don’t understand how zetapotential can be measured if the sample contains no nano-objects.

Page 12 L.278. It is not possible to assign specifically the 350 to a transistion, as it can also originate from π- π* of conjugated system formed during the cdot generation.

For fluorescence measurement, the authors should state that fluorescence measurement were done with absorbance below 0,1 in order to avoid inner filter effect. If the fluorescence was not measured with this experimental condition, it can explain the change of the wavelength of emission peaks. I guess that the fluorescence was not measured directly with samples presented in Fig 2 B,C,D as they seem too much concentrated for fluorescence measurements.

Page 14.L.312 I don’t know how the value (40%) of the singlet oxygen quantum is measured.

Page 16. Figure 5 B. In this figure Cdot and PPIX are compared. However, it is impossible for the reader to be sure of the validity of this experiment as the absorption of the samples is not given. PPIX singlet oxygen quantum is reported to be 0.4 (Photochem. Photobiol. 1999, 70 (4), 391–475) in water, that would means, according to figure 5 B, that Cdot1:4 quantum yield is quantitative.   

Author Response

Attached file.

Reviewer 4 Report

The authors present an important work about carbon dots associated with doxorubicin as a photodynamic anti-cancer therapy.

They present a large amount of physico-chemical data on these carbon dots as well as in vitro test results.  I look forward to seeing the results in vivo on solid tumours.

Some minor remarks:

Line 43: In the introduction: the authors say that cancers has become the most common cause of death…  but in fact cancers are the second cause of death and cardiovascular disease is the leading cause of death in the world (according to the WHO). Can the authors modify and add a reference?

Ref 2 and 3 are not the best, not very specific, can the author change for more appropriate references (reviews)?

Ref 4: The authors say that there is a synergistic effect of PDT and chemotherapy, but the cited article (Ref 4) does not mention PDT. Please check your reference and add an appropriate reference.

Lines 136 and 144:  Please write 6kDa

Lines 166 and 172: Why is the irradiation time at 450 nm so long?

Figure 3: Aa and Ab: please have a similar x-axis; B: Please indicate in the figure and/or in the legend the % of water in each tube; C: indicate the % of amine in each tube.

Fig S3: Please have the absorbance scale similar for figures c and a (for better comparison).

Fig 6c: Please change the x-axis: 0 to 100

Fig 6 Da and Db: change the minutes to hours on the y-axis

Lines 368-369: the sentence is not clear, please rephrase

Lines 440 and 441: Why is it more sustainable, the release is less than that of other groups. Can you explain this please?

Author Response

Attached file.

Round 2

Author Response

attached file.

Round 3
